# Autonomous metabolic reprogramming and oxidative stress characterize endothelial dysfunction in acute myocardial infarction

Erika Zodda[1,2,3], Olga Tura-Ceide[4,5,6], Nicholas L Mills[7], Josep Tarragó-Celada[1], Marina Carini[8], Timothy M Thomson[2,3,9]*[†], Marta Cascante[1,3,10]*[†]

[1]Department of Biochemistry and Molecular Biology, Faculty of Biology, University of Barcelona, Barcelona, Spain; [2]Institute for Molecular Biology of Barcelona, National Research Council (IBMB-CSIC), Barcelona, Spain; [3]Centro de Investigación Biomédica en Red de Enfermedades Hepáticas y Digestivas (CIBER-EDH), Madrid, Spain; [4]Department of Pulmonary Medicine, Hospital Clínic-Institut d'Investigacions Biomèdiques August Pi I Sunyer (IDIBAPS); University of Barcelona, Barcelona, Spain; [5]Centro de Investigación Biomédica en Red de Enfermedades Respiratorias (CIBERES), Madrid, Spain; [6]Department of Pulmonary Medicine, Dr. Josep Trueta University Hospital de Girona, Santa Caterina Hospital de Salt and Girona Biomedical Research Institute (IDIBGI), Girona, Spain; [7]University/BHF Centre for Cardiovascular Science, University of Edinburgh, Edinburgh, United Kingdom; [8]Department of Pharmaceutical Sciences, Università degli Studi di Milano, Milan, Italy; [9]Universidad Peruana Cayetano Heredia, Lima, Peru; [10]Institute of Biomedicine (IBUB), University of Barcelona, Barcelona, Spain

*For correspondence:
titbmc@ibmb.csic.es (TMT);
martacascante@ub.edu (MC)

[†]These authors contributed equally to this work

Competing interest: The authors declare that no competing interests exist.

**Abstract** Compelling evidence has accumulated on the role of oxidative stress on the endothelial cell (EC) dysfunction in acute coronary syndrome. Unveiling the underlying metabolic determinants has been hampered by the scarcity of appropriate cell models to address cell-autonomous mechanisms of EC dysfunction. We have generated endothelial cells derived from thrombectomy specimens from patients affected with acute myocardial infarction (AMI) and conducted phenotypical and metabolic characterizations. AMI-derived endothelial cells (AMIECs) display impaired growth, migration, and tubulogenesis. Metabolically, AMIECs displayed augmented ROS and glutathione intracellular content, with a diminished glucose consumption coupled to high lactate production. In AMIECs, while PFKFB3 protein levels of were downregulated, PFKFB4 levels were upregulated, suggesting a shunting of glycolysis towards the pentose phosphate pathway, supported by upregulation of G6PD. Furthermore, the glutaminolytic enzyme GLS was upregulated in AMIECs, providing an explanation for the increase in glutathione content. Finally, AMIECs displayed a significantly higher mitochondrial membrane potential than control ECs, which, together with high ROS levels, suggests a coupled mitochondrial activity. We suggest that high mitochondrial proton coupling underlies the high production of ROS, balanced by PPP- and glutaminolysis-driven synthesis of glutathione, as a primary, cell-autonomous abnormality driving EC dysfunction in AMI.

## Editor's evaluation

In a novel and important approach, the authors have generated coronary artery endothelial cells from thrombectomy specimens of patients with acute myocardial infarctions (AMIECs). Compared to

control healthy coronary artery endothelial cells (HCAECs) they display solid evidence for impaired growth, migration and tubulogenesis and major metabolic differences. This advances the field and provides opportunities for future studies including investigating new avenues of therapy for AMI.

## Introduction

Cardiovascular diseases are the leading cause of death worldwide and comprise a range of syndromes affecting the functionality of the heart and blood vessels (*Smolders et al., 2018*; *Townsend et al., 2022*). Vascular dysfunction is believed to be a primary factor in the onset and progression of atherosclerosis and other vascular-related disorders. In this study, we focus our attention on a specific vascular disease, AMI, a major cause of morbidity in the Western world (*Townsend et al., 2022*). AMI occurs as a consequence of coronary occlusion, generally from a thrombus superimposed on an ulcerated or unstable atherosclerotic plaque (*Badimon et al., 2012*; *Smolders et al., 2018*). The endothelium plays a pivotal role in the development of AMI, both preceding atherogenesis and predisposing to thrombosis (*Previtali et al., 2011*).

Emerging evidence indicates that pathological blood vessel responses and endothelial dysfunction are associated with metabolic alterations in ECs (*Botts et al., 2021*; *Moreno-Viedma et al., 2016*). Thus, to identify new strategies to limit CVD progression in at risk/affected patients, metabolic alterations underlying impaired endothelial cells functions require appropriate patient-derived cell models (*Gallogly et al., 2021*). As such, the study of endothelial metabolism and mitochondrial function may be central to unveiling fundamental mechanisms of cardiovascular pathogenesis, and to identify novel critical metabolic biomarkers and therapeutic targets.

Here, we have approached the study of endothelial cells dysfunction and metabolic alterations associated with AMI, by using coronary endothelial cells isolated and cultured from thrombectomy specimens obtained from patients affected with AMI. Our observations point to pharmacologically actionable cell-autonomous metabolic dysfunction in these cells.

## Results

### AMIECs display impaired growth and migration

AMIECs were isolated from coronary atherothrombotic specimens from eight patients undergoing percutaneous coronary intervention with thrombectomy for the treatment of acute ST-segment elevation myocardial infarction (STEMI). Hereafter, the eight patient-derived cell lines are designated CE followed by two digits (CE45, CE46, CE48, CE51, CE52, CE53, CE60, CE61).

At 80% confluence, AMIECs presented the expected shape and appeared similar to control HCAECs (*Figure 1A*), as expected for bonafide endothelial cells grown under equivalent conditions. The endothelial nature of these cells was confirmed by Western blotting for the expression of the endothelial markers, V-CAM1 (vascular cell adhesion protein 1, CD-106), PECAM-1 (platelet endothelial cell adhesion molecule, CD31), CD-34, and CD-105 (endoglin) (*Figure 1B and C*).

Because optimal growth conditions for endothelial cells entail maintaining them at a relatively high confluence, they cannot be kept in culture for prolonged periods at an optimal confluence (70–80%), as overgrowth will cause cells to pile up and lose their physiological monolayer growth pattern. Therefore, cell growth monitoring was limited to 168 hr. Under these conditions, all AMIECs consistently displayed slower growth rates as compared to control HCAECs (*Figure 1D*). Indeed, after 168 hr, while control HCAECs had reached a plateau in their growth, AMIECs continued to proliferate exponentially, albeit at slow rates, without reaching saturation.

Given these results, we determine the cell cycle distribution of these cells by flow cytometry 24 hr after seeding, the point of maximum growth in the control HCAECs. Somewhat contrary to our expectations, we observed no significant differences in the distribution of cell cycle phases between AMIECs and control cells (*Figure 1E*), except for a modest tendency, without reaching statistical significance, of AMIECs to accumulate in the G1 phase relative to HCAECs. The slow growth rate, in conjunction with unaltered cell cycle phase distribution, of AMIECs suggests an overall slowing down of growth that affects equally all phases of the cell cycle.

In wound-healing migration assays, control HCAECs achieved a complete closure of the wound after 24 hr of seeding (*Figure 1F*). In contrast, three AMIECs displayed wound closure rates below

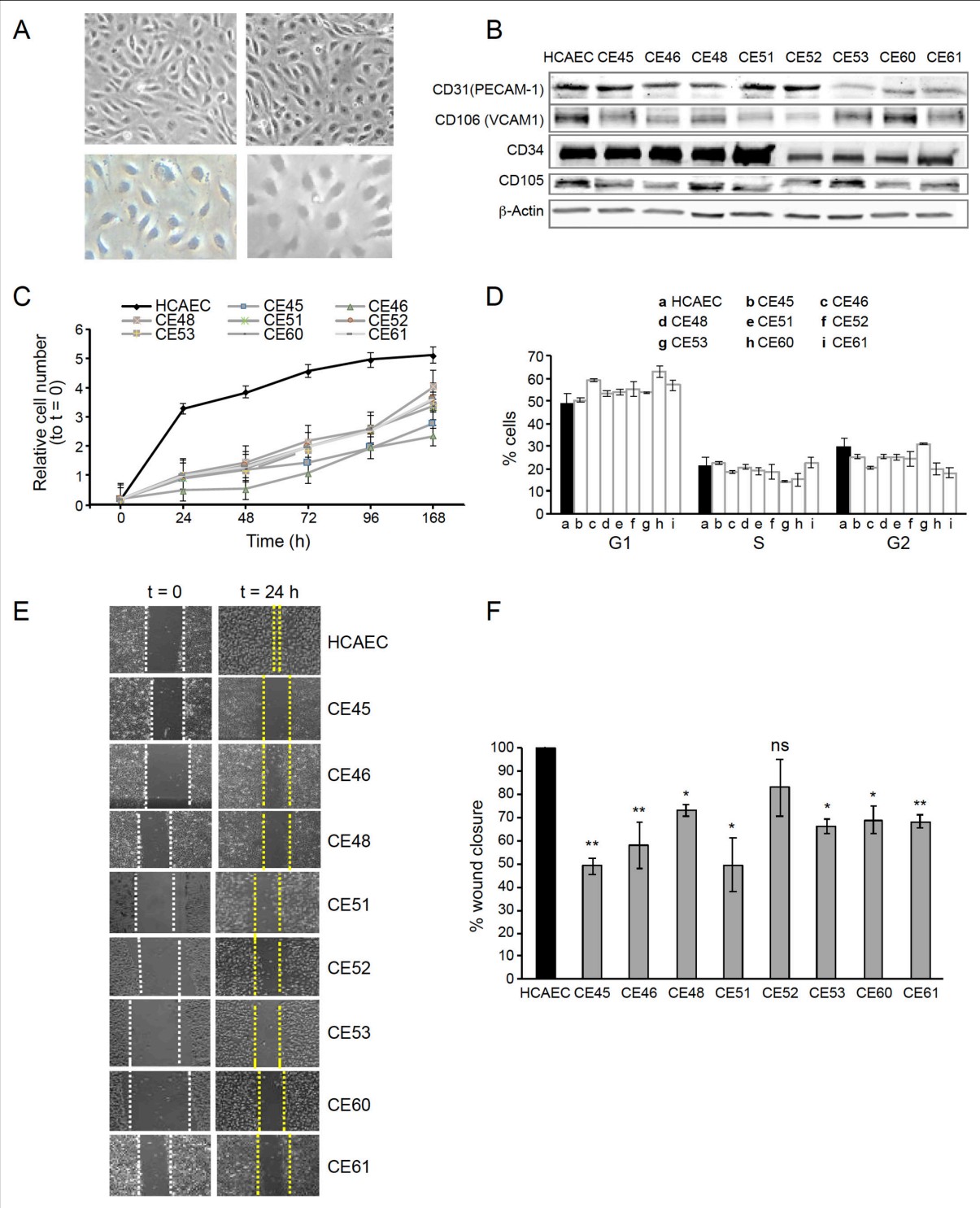

**Figure 1.** AMI-derived endothelial cells (AMIECs) display impaired growth and migration. (**A**) Bright field images of human coronary artery endothelial cells (HCAECs) (left panels) vs. AMIECs (right panels), captured under 10 x and 20 x objectives on a Leica microscope. (**B**) Representative Western blotting experiments for the endothelial markers, PECAM-1, V-CAM1, CD-34, and CD-10 (*Figure 1—source data 1 and 2*). (**C**) Growth rate of AMIECs and control HCAEC, expressed as fold change relative to initial cell number seeded. Error bars represent mean ± SD (n=3). (**D**) Cell cycle analysis of AMIECs and HCAECs under logarithmic growth conditions, showing a non-significant trend for AMIECs to accumulate G1 phase as compared to HCAEC. Error bars represent mean ± SD (n=3). (**E**) Representative images of Mito C-treated migration in scratch wound assays, showing reduced migration of AMIECs compared to control HCAECs. (**F**) Quantification of cell migration in scratch wound assays, represented as % of wound closure. Values are mean ± SD (n=3). Student's t-test significance values were calculated for AMIECs vs. HCAEC: *p≤0.05 **p≤0.01, ***p≤0.001.

*Figure 1 continued on next page*

*Figure 1 continued*

The online version of this article includes the following source data for figure 1:

**Source data 1.** Western blotting experiments for endothelial markers.

**Source data 2.** Western blotting experiments for endothelial markers.

50%, four others about 60%, and only one about 80% after 24 hr of migration under the same conditions (*Figure 1G*).

These results indicate that AMI patient-derived coronary endothelial cells are dysfunctional in major cell-autonomous processes, namely growth and migration.

## Defective tubulogenesis in AMIECs

The ability to migrate is a critical function of healthy endothelial cells, which can migrate while branching from existing blood vessels toward sources of angiogenic stimuli, guided by tip cells. Impairment in

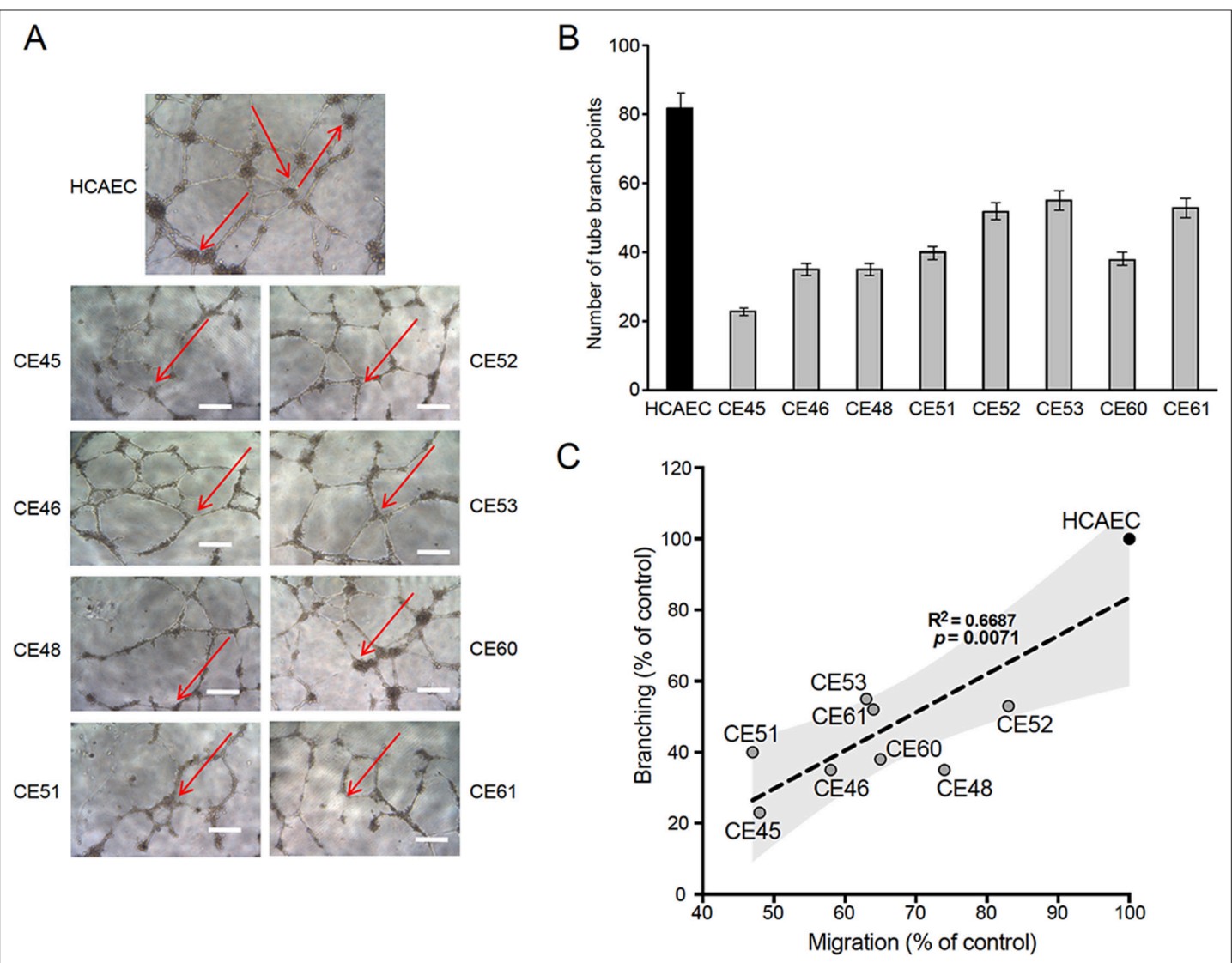

**Figure 2.** Defective tubulogenesis of AMI-derived endothelial cells (AMIECs). (**A**) Endothelial tube network formation after 12 hr of VEGF stimulation. Shown are representative bright field images from triplicate experiments. Scale bar = 100 μm. (**B**) Tube branch point quantification of images from experiments performed as in (**A**). Values are mean ± SD (n=3). Student's t-test significance values were calculated for AMIECs vs. human coronary artery endothelial cell (HCAEC): *p≤0.05 **p≤0.01, ***p≤0.001. (**C**) Correlation between relative migration (*Figure 1F*) and tubulogenesis in AMIECs and HCAECs.

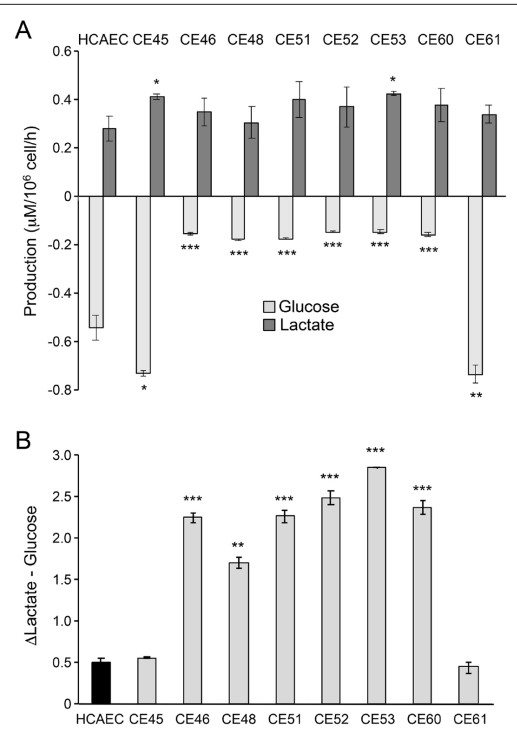

**Figure 3.** AMI-derived endothelial cells (AMIECs) display low glucose consumption and high lactate production. (**A**) Glucose consumption and Lactate production have been measured after 48 hr. (**B**) Conversion rate of glucose to lactate. Error bars represent mean ± SD (n=3). Student's t-test significance values were calculated for AMIECs vs. Human coronary artery endothelial cell (HCAECs): *p≤0.05 **p≤0.01, and ***p≤0.001.

this fundamental endothelial function is considered a hallmark of vascular diseases (*De Bock et al., 2013*). Thus, we next assessed the tubulogenic potential of AMIECs, as an indicator of their vasculogenic (angiogenic) potential. Control HCAECs and AMIECs were cultured under conditions that favor tubulogenesis, and branching points scored after 12 hr of switching to a tubulogenic medium containing VEGF (*Figure 2A*). While control HCAECs averaged 80 branching points, AMIECs displayed a significantly reduced tubulogenic potential, evident both as a diminished number of branching points and as shorter and fragmented tubes as compared to control cells (*Figure 2B*). The branching capacity of the AMIECs derived from different patients was linearly correlated ($R^2$=0.67) with their ability to migrate (*Figure 2C*) regardless of their growth rates suggesting that the deficient tubulogenesis or sprouting of these cells is linked to a defect in their migration properties, rather than to abnormal proliferation or survival.

## AMIECs display low glucose consumption and high lactate production

The above characterization points to cell-autonomous defects in AMI patient-derived endothelial cells that compromise their ability to proliferate, migrate, and form branched tubular structures. We next tackled the hypothesis that the metabolic features of these cells might be associated with at least part of the observed phenotypes. To assess nutrient and metabolite consumption and production rates, the concentration of selected metabolites in culture media was determined. We focused on glucose as the major carbon source of the glycolytic pathway, lactate as the main metabolic end-product of glycolysis, and glutamine/glutamate as alternative sources of carbon metabolism and major providers of mitochondrial anaplerosis. We found that 6 out of the 8 AMIECs consumed glucose at significantly lower rates than control HCAECs (*Figure 3A*), which may reconcile with the lower growth rates of patient-derived cells, while two patient-derived cell lines consumed more glucose than control cells. Remarkably, all AMIECs displayed a strong production of lactate, comparable or superior to control cells, regardless of their levels of glucose uptake (*Figure 3B*).

Two of the AMIECs, CE-45, and CE-61, consumed high levels of glucose and displayed robust lactate production (*Figure 3A and B*), and thus, it may be assumed that lactate is produced through glycolysis. The cells may also be hypothesized to be under a Warburg effect. However, their slow rates of proliferation stand in contrast to the rapid proliferation phenotypes seen in other cell types displaying a Warburg effect, such as cancer cells. On the other hand, the remaining six patient-derived endothelial cell lines, also slow growers, showed high levels of lactate production in spite of low glucose consumption (*Figure 3A and B*). This uncoupling of lactate production from glycolysis suggests the existence of an alternative source of lactate in these cells. In this regard, it is worth noting that glutaminolysis also produces lactate, in a pathway that involves the oxidation of glutamine to malate and then to pyruvate by the malic enzyme (ME), followed by conversion to lactate by the action of LDH-A (*Brooks, 2018*).

## AMIECs shunt glycolysis towards PPP and NADPH generation

The low glucose uptake observed in 6 out of the 8 patient-derived endothelial cell lines prompted us to explore the expression of key enzymes in the glycolytic pathway. Under normal conditions, endothelial cells rely mainly on glycolysis to generate the energy required for their functions (*De Bock et al., 2013*; *Dromparis and Michelakis, 2013*; *Goveia et al., 2014*). A key rate-limiting enzyme in this pathway is phosphofructokinase 1 (PFK1), responsible for the phosphorylation of fructose-6-phosphate (F-6-P) to fructose-1,6-phosphate (F-1,6-P) in an ATP-dependent reaction. PFK1 is allosterically activated by fructose-2,6-bisphosphate (F-2,6-BP), produced from F-6-P through the catalytic activity of 6-phosphofructo-2-kinase/fructose-2,6-bisphosphatases encoded in mammals by paralogs PFKFB1,

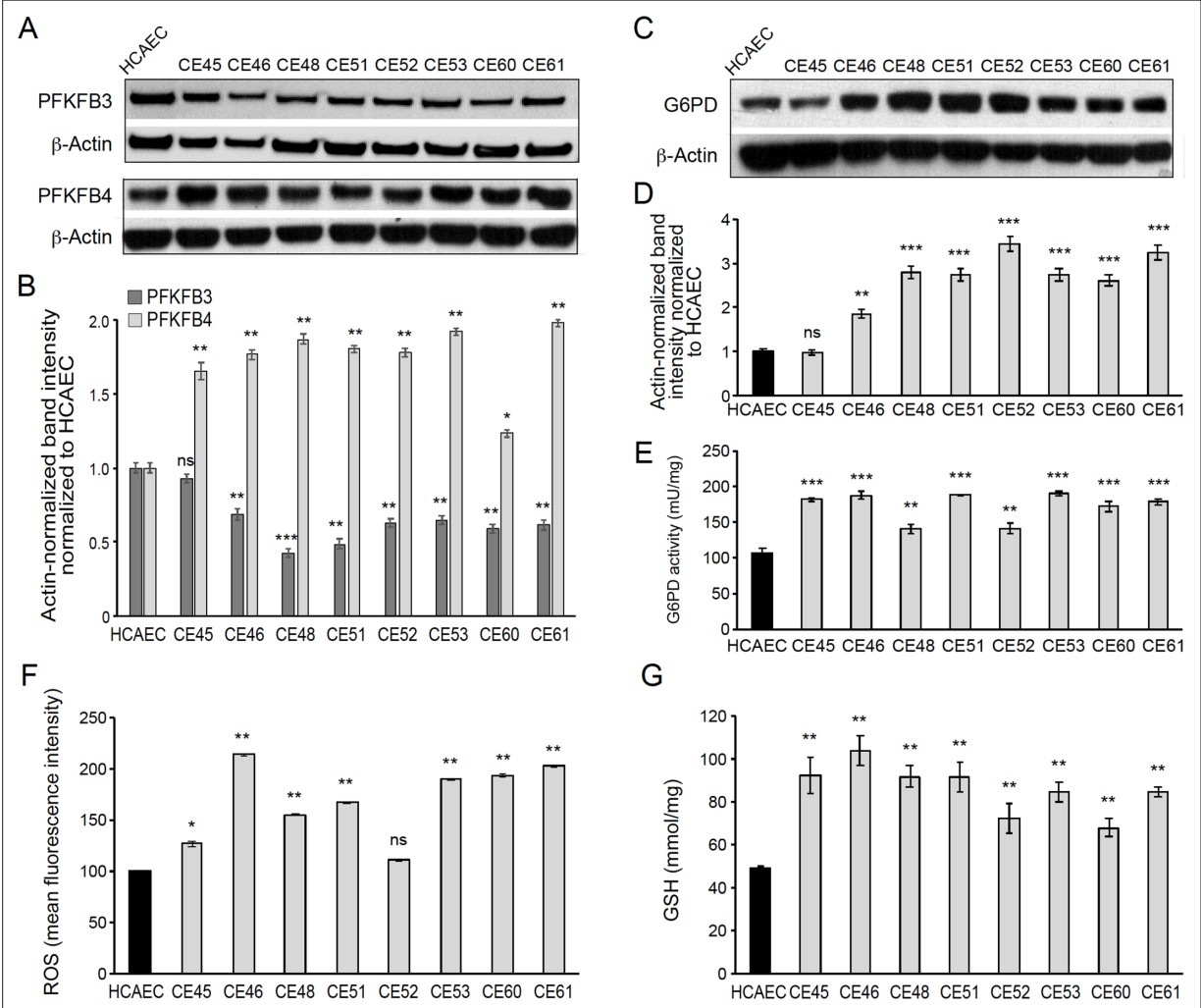

**Figure 4.** AMI-derived endothelial cells (AMIECs) shunt glycolysis towards pentose phosphate pathway (PPP) and NADPH generation. (**A**) Representative Western blotting of PFKFB3 and PFKFB4 in AMIECs and human coronary artery endothelial cells (HCAECs) (*Figure 4—source data 1* ). (**B**) Quantification of band intensities for PFKFB3 and PFKFB4 in triplicate western blotting experiments, normalized to β-actin band intensities. Values are mean ± SD (n=3). (**C**) Representative Western blotting for glucose-6-phosphate dehydrogenase (G6PD) in AMIECs and HCAECs (*Figure 4—source data 2*). (**D**) Quantification of band intensities for G6PD in triplicate Western blotting experiments, normalized to β-actin band intensities. (**E**) G6PD enzyme activities normalized to intracellular protein content (n=3). (**F**) Reactive oxygen species (ROS) levels are determined by flow cytometry. Results are expressed as the mean fluorescent intensity. (**G**) Total intracellular reduced glutathione content. Values are normalized to cellular protein concentration. Error bars represent mean ± SD (n=3). For all experiments, Student's t-test significance values were calculated for AMIECs vs. HCAECs: *p≤0.05 **p≤0.01, and ***p≤0.001.

The online version of this article includes the following source data for figure 4:

**Source data 1.** Western blotting of PFKFB3 and PFKFB4 in AMIECs and HCAECs.

**Source data 2.** Western blotting for G6PD in AMIECs and HCAECs.

PFKFB2, PFKFB3, PFKFB4 and TP53-induced glycolysis and apoptosis regulator (TIGAR) (*Bensaad et al., 2006*; *Eelen et al., 2018*). These are bifunctional enzymes bearing, on the same polypeptide, an N-terminal 6-phosphofructo-2-kinase domain, and a C-terminal fructose 2,6-bisphosphatase domain. Given its relative kinase and bisphosphatase activities, PFKFB3 is a preferred target to inhibit glycolysis over other isoforms in this family of enzymes. PFKFB3 has been described to play a significant role in the glycolytic activity of pathological ECs, and its inhibition dramatically abrogates their proliferation and migration (*Li et al., 2019*; *Schoors et al., 2014*).

We determined PFKFB3 protein levels by Western blotting, finding significantly lower levels in 7 out of the 8 AMIECs, as compared to control HCAECs (*Figure 4A and B*), consistent with their lower glucose uptake. In fact, one of the two patient-derived cell lines with high glucose uptake, CE-45, expressed PFKFB3 at levels close to those of control cells (*Figure 4A and B*), thus reinforcing the notion that cell lines with low glucose uptake have low rates of glycolysis. This may also help explain, at least in part, the low proliferation rates, migration capacity, and tubulogenic activity of these cells, as these processes require robust glycolysis.

We also determined PFKFB4 protein levels, finding significantly higher levels of expression in AMIECs compared to control cells (*Figure 4A and B*). Low levels of PFKFB3 expression coupled with high levels of PFKFB4 expression in AMI patient-derived cells predict a redirection of glucose-6-phosphate (G6P) towards the PPP. As a consequence, we would expect (1) increased production of NADPH-reducing equivalents through the oxidative branch of the PPP (*Dasgupta et al., 2018*; *Ros et al., 2017*) and (2) increased production of ribose-5-phosphate through the non-oxidative branch, for de novo synthesis of nucleotides as well as acetyl-CoA carboxylase 1 activation and lipogenesis (by inhibiting activation of liver kinase B1/AMPK signaling). The NADPH-reducing equivalents produced through the non-oxidative branch afford the maintenance of redox homeostasis by scavenging ROS (*Benito et al., 2017*; *Li et al., 2014*; *Sun et al., 2017*; *Xia et al., 2017*). The nucleotides supplied through the non-oxidative branch of the PPP can be used for DNA synthesis in proliferating cells, and also in DNA damage repair (*Bester et al., 2011*).

In light of these predictions, we determined the expression level and activity of glucose-6-phosphate dehydrogenase (G6PD), the rate-limiting enzyme of the oxidative branch of the PPP. Consistently, we found significantly higher levels of G6PD in AMIECs than in control cells (*Figure 4C and D*). Likewise, AMIECs displayed higher levels of G6PD enzyme activity than control cells (*Figure 4E*). These results support an enhanced activity of the oxidative branch of the PPP. This is expected to generate higher levels of NADPH, which, in turn, should contribute to improving ROS scavenging by maintaining glutathione in its reduced state. We thus determined ROS and glutathione levels in our cell model. Seven out of the 8 AMIECs contained ROS at significantly higher levels than control cells (*Figure 4F*). The one cell line with near-normal ROS levels, CE-45, was the same one displayed normal levels of G6PD protein (*Figure 4C and D*). Similarly, the intracellular concentration of reduced glutathione was increased in all 8 AMIECs (*Figure 4G*). These observations suggest that glutathione levels are upregulated in AMIECs to counter excess ROS levels.

In vivo, endothelial cells in AMI patients are under numerous physiological and pathological stresses, including mechanical and inflammatory, which may generate excess ROS and oxidative stress (*Dasgupta et al., 2018*; *Seo and Lee, 2014*). However, our patient-derived cells are no longer in an in vivo environment and are rather in an in vitro environment identical to that of control endothelial cells, and, therefore, their increased ROS levels cannot be directly attributed to environmental factors. Absent maintenance of long-term epigenetic memory of past stress responses in vivo, it is reasonable to assume that the high levels of ROS in AIM patient-derived cells may be elicited by endogenous damage signals impinging upon mitochondria (*Murphy, 2009*) or through NAD(P)H oxidases.

## AMIECs exhibit a strong glutamine metabolism

Given the likely relevance of glutathione in the observed phenotypes and the possibility of a glutamine origin of the excess lactate produced in AMIECs, we assessed glutamate and cysteine contents in these cells. Determination of the extracellular fluxes of glutamine showed that the consumption of glutamine in AMIECs is higher than in control HCAECs (*Figure 5A*). The latter consumed more glucose than patient-derived cells (see above), and thus the glutamine-to-glucose consumption ratio is far greater in patient-derived cells than in control cells (*Figure 5B*). This suggests that AMI patient-derived cells mainly resort to glutamine, rather than glucose, as their carbon source.

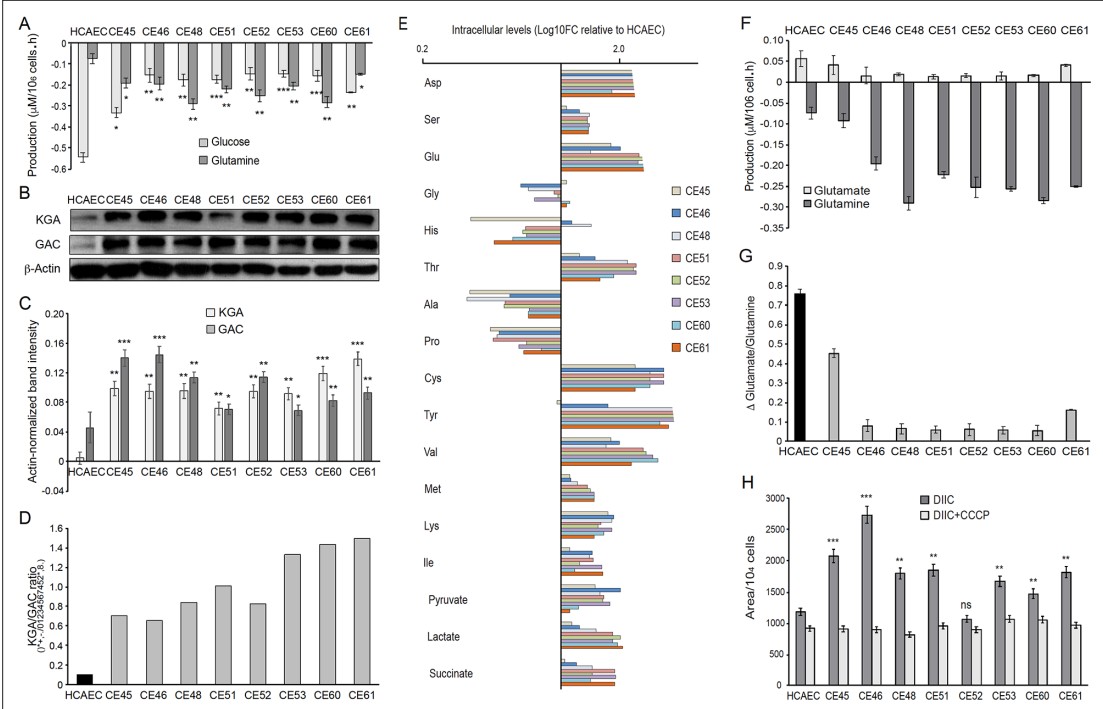

**Figure 5.** AMI-derived endothelial cells (AMIECs) display a strong glutamine metabolism. (**A**) Glutamine and glucose production levels in cell culture medium after 48 hr. Negative values reflect net consumption. (**B**) Representative Western blotting experiments in AMIECs and human coronary artery endothelial cells (HCAECs) for the two major isoforms of GLS1, kidney-type glutaminase (KGA), and glutaminase C (GAC), using isoform-specific antibodies (*Figure 5—source data 1*). (**C**) Quantification of band intensities for KGA and GAC in triplicate western blotting experiments, normalized to β-actin band intensities. (**D**) Ratio of KGA vs. GAC isoform expression levels, calculated from the actin-normalized band intensities quantitated in (**C**). (**E**) Intracellular amino acid levels (**F**) Glutamate and glutamine production levels in cell culture medium after 48 hr. Negative values reflect net consumption. (**G**) Glutamine/glutamate consumption ratio. (**H**) Mitochondrial metabolic potential of AMIECs and control HCAEC cells. Cells were stained with 50 nM DiIC$_1$(5) without or with 50 μM carbonyl cyanide 3-chlorophenylhydrazone (CCCP). Error bars represent mean ± SD (n=3). For all experiments, Student's t-test significance values were calculated for AMIECs vs. HCAECs: *p≤0.05 **p≤0.01, and ***p≤0.001.

The online version of this article includes the following source data for figure 5:

**Source data 1.** Western blotting experiments in AMIECs and HCAECs for the two major isoforms of GLS1, KGA and GAC.

Glutamine is the most abundant free amino acid in circulation and in intracellular pools, and it is used by cells for both bioenergetics and biosynthetic needs; it can act as a carbon source or nitrogen donor and is consequently used as a precursor for the synthesis of amino acids, proteins, nucleotides and lipids, and represents the main substrate for TCA cycle anaplerosis (*DeBerardinis and Cheng, 2010*; *Draoui et al., 2017*; *Fitzgerald et al., 2018*; *Laplante and Sabatini, 2012*). Moreover, glutamine participates in redox homeostasis, a process that has a close connection with the PPP, as one of its main functions is to generate glutathione to counter the oxidative stress generated by ROS (see above). Thus, the increased glutamine consumption observed in AMIECs is consistent with the increased glutathione levels observed in these cells (*Figure 4G*).

The anaplerotic role of glutamine requires its conversion to glutamate, in a reaction catalyzed by glutaminase (GLS). There are three isoforms of human GLS: the kidney-type (KGA) and its splice variant glutaminase C (GAC) encoded by the GLS1 gene, and the liver-type (LGA) encoded by the GLS2 gene (*Thangavelu et al., 2014*; *van den Heuvel et al., 2012*). Notably, the GAC isoform is crucial for cell growth under acidic pH, as would be expected in the presence of high levels of lactate (*Aguilar et al., 2016*). Given the high levels of glutamine consumption observed in patient-derived cells, we determined the expression of glutaminase by Western blotting, using isoform-specific antibodies (*Figure 5B*). Consistently, both isoforms of GLS1, KGA, and GAC, are expressed at significantly higher levels in patient-derived cells than control cells (*Figure 5B and C*), with a higher KGA/GAC ratio in AMIECs (*Figure 5D*).

Next, polar intracellular metabolites (TCA cycle intermediates and amino acids) were determined by GC/MS. The results disclosed a higher intracellular content of all amino acids determined, with the exception of glycine, alanine, histidine, and proline. Among AMIECs, CE45 displayed the intracellular amino acid profile most resembling that of control HCAECs (*Figure 5E*), consistent with the overall normal-like phenotype of this cell line. Cysteine and glutamate play important roles in the synthesis of glutathione (GSH) (*Ishimoto et al., 2011*) and both had significantly higher concentrations in all AMIECs (*Figure 5E*), consistent with higher levels of GSH as compared to control cells. Most of the glutamine transported into the cell is converted to glutamate by glutaminase, which is then converted to α-ketoglutarate (αKG) by glutamate dehydrogenase (GLDH). αKG is used as a carbon source for the anaplerotic replenishment of the TCA cycle, while glutamate, which normally accumulates in the cells, is a precursor of GSH. Thus, glutamate also contributes to redox homeostasis and ROS detoxification, along with the PPP. We thus determined glutamate levels in culture media, finding lower levels of production in 6 out of the 8 patient-derived cells as compared to control cells (*Figure 5F*). Moreover, the ratio between glutamate production and glutamine consumption was considerably lower in AMIECs (*Figure 5G*). Because of the high levels of glutaminase in patient-derived cells, implying a strong conversion of glutamine to glutamate, the lower levels of extracellular export of glutamate are suggestive of a robust consumption of glutamate in the production of GSH in these cells.

Collectively, our data suggest that AMIECs rely heavily on glutamine, both as an alternative to glucose as a carbon source and in order to maintain redox homeostasis in these cells. Related to our observations, a link between glutamine metabolism and clinical manifestations of atherosclerosis, such as intima-media thickness and plaque development, has been described (*Grajeda-Iglesias and Aviram, 2018*; *Wurtz et al., 2012*).

These observations are consistent with higher levels of mitochondrial respiration and higher anaplerotic use of glutamate. The transport of glutamate into mitochondria is also essential for the biosynthesis of aspartate (*Wong et al., 2016*). Consistently, aspartate was found at increased levels in AMIECs (*Figure 5E*). Aspartate derived from oxaloacetate, via glutaminolysis, produces NAD+, required for glycolysis and NADPH, needed for redox balance. Aspartate derived from oxaloacetate, via glutaminolysis, produces NAD+, required for glycolysis and NADPH, needed for redox balance (*Wong et al., 2016*). Therefore, this pathway also contributes to offset the high levels of ROS in these cells.

A possible source of the high levels of ROS observed in patient-derived cells is increased mitochondrial respiration in these cells. On the other hand, high levels of glutaminase and glutamine consumption suggest a strong activation of glutaminolysis in patient-derived cells. Once in the internal space of the mitochondria, glutamate is converted into α-ketoglutarate and ammonia by glutamate dehydrogenase, along with the reduction of $NAD(P)^+$ into NADPH. NADH is the substrate of complex I of the mitochondrial respiratory chain, and its oxidation results in the translocation of $H^+$ across the membrane, thus producing a proton gradient. Together, our evidence predicts an increased mitochondrial membrane potential in these cells. High membrane potential implies large negative potentials in the mitochondrial matrix, leading to glutamate expulsion and high glutaminase activity. Maintenance of the mitochondrial membrane potential ($\Delta \psi$ m) is an essential component in the process of energy storage during oxidative phosphorylation, as it reflects the pumping of $H^+$ across the inner membrane during the process of electron transport and oxidative phosphorylation (*Zorova et al., 2018*).

We thus determined the status of the mitochondrial membrane potential in our cells. For this, we used the fluorescent probe DiIC1(5) (MitoProbe), a positively charged, cell-permeant dye that accumulates within the negatively-charged mitochondrial matrix. MitoProbe accumulated at significantly higher levels in all AMIECs relative to control HCAECs (*Figure 5H*), indicating a mitochondrial hyperpolarization due to increased membrane potentials. The subsequent addition of the uncoupling agent carbonyl cyanide m-chlorophenylhydrazone (CCCP) resulted in rapid depolarization, reflecting the proton gradient dissipation.

## Discussion

Understanding metabolic reprogramming as a paramount process driving the endothelial dysfunction that underlies atherogenic cardiovascular diseases is critical for designing new therapeutic approaches, in conjunction with conventional interventions, to tackle still unmet needs for better clinical management of these broad categories of disease. Much of the accrued evidence for metabolic

and mitochondrial dysfunction in ECs in atherogenesis comes from animal models (*Negre-Salvayre et al., 2020*) or studies in patients (*Shemiakova et al., 2020*), critical scenarios to achieve a contextual understanding of the pathophysiological processes at stake.

In this study, we have departed from those approaches, in an attempt to better delineate cell-autonomous metabolic features of ECs derived from AMI patients, or AMIECs. We have found that contrary to physiologically normal ECs, which rely mostly on glycolysis for energy production rather than mitochondrial respiration (*Krützfeldt et al., 1990*), AMIECs display low levels of glycolysis, as attested by low glucose consumption and downregulation of PFKFB3, which catalyzes the production of fructose-2,6-bisP (F2,6BP), a potent allosteric activator of the glycolysis rate-limiting enzyme 6-phosphofructokinase-1 (PFK-1), while also displaying a high membrane potential, suggestive of a high mitochondrial activity. Consistent with high levels of mitochondrial activity, AMIECs display a high intracellular content of ROS. The combination of high membrane potential and high ROS levels suggests robust proton coupling at the inner mitochondrial membrane, as uncoupling through the expression of UCP1 or the use of pharmacological uncouplers has been shown to mitigate ROS production (*Cadenas, 2018*). We have found that the augmented ROS content is counteracted in AMIECs by an augmented intracellular content of glutathione, a physiological homeostatic mechanism to prevent damage from excess ROS (*Hayes et al., 2020*). We also provide evidence that the major pathways enabling the biosynthesis of glutathione, the PPP and glutaminolysis, are overactivated in AIMECs as compared to control coronary ECs. The relatively low levels of expression of PFKFB3 and high levels of PFKFB4 point to a shunting of glycolysis towards the PPP in AMIECs, in order to produce NADPH.

Both PFKFB4 and PFKFB3 are upregulated in hypoxia through transcriptional activation by HIF-1α (*Minchenko et al., 2014*), the mTOR pathway (*Feng and Wu, 2017*), and peroxisome proliferator-activating receptor γ (PPARγ) (*Shu et al., 2016*). On the other hand, although wild-type p53, activated in response to DNA damage, transcriptionally represses both PFKFB3 and PFKFB4, PFKFB4, but not PFKFB3, has been described as strongly upregulated in the absence of p53 (*Ros et al., 2017*).

While, similar to HIF-1α, mTOR, and PPARγ drive glycolysis, our patient-derived endothelial cells show evidence of diminished glycolysis accompanied with low expression levels of PFKFB3, and thus it seems unlikely that the high levels of PFKFB4 observed in these cells are a downstream outcome of the activation of these pathways. The hypothesis of a blunted function of p53 as a driver of PFKFB4 upregulation in these cells could merit further investigation. Interestingly, the malic enzyme isoforms, ME1 and ME2, are also under negative regulation by p53 and are upregulated in the absence of p53 (*Ros et al., 2017*). Regardless of the mechanisms underlying PFKFB3 low-expression along with PFKFB4 over-expression in AMIECs, predicted downstream consequences are (1) decreased levels of F-2,6-P2 resulting in low PFK1 activity and diminished glycolytic flux, which is consistent with the observed phenotypes, and (2) shunting of G6P towards the PPP, reinforced by the activation by PFKFB4 of steroid receptor coactivator-3 (SRC-3), enhancing its transcriptional activity and stimulating the expression of transketolase (TKT) (*Dasgupta et al., 2018*).

The high levels of lactate production observed in AMIECs, concomitant with low glucose consumption, represent a conundrum. High lactate production is frequently linked to strong glycolytic activity, as the glycolytic end-product, pyruvate, can be converted to lactate catalyzed by the enzymatic activity of lactate dehydrogenase-A (LDH-A). Indeed, in highly proliferating cells, such as cancer or progenitor cells, a high rate of glycolysis can be accompanied by an inhibition of the mitochondrial entry of pyruvate, which, instead, is diverted towards the generation of lactate, a metabolic rewiring designated the Warburg effect, or aerobic glycolysis (*Burns and Manda, 2017*). Strong glycolysis coupled with high lactate production and mitochondrial shutdown is also characteristic of hypoxia, through transcriptional regulation by HIF-1α of key regulators of these three processes, and has been observed in dysfunctional endothelial cells associated with atherosclerosis and vascular diseases (*Fitzgerald et al., 2018*; *Theodorou and Boon, 2018*).

In the LDH-A catalyzed reaction, pyruvate accepts electrons from NADH, yielding lactate and replenishing NAD$^+$, predicted to be depleted in AMIECs by a strong drive to produce reducing equivalents and by the increased mitochondrial membrane potential, which impairs electron transport and NAD + regeneration (*Luengo et al., 2021*). However, under conditions of low glucose consumption and evidence of low glycolytic activity, an alternative source is needed that may explain high lactate production from pyruvate in AMIECs. One possibility is through glutamine-initiated metabolism

(*Zielke et al., 1980*), which would be consistent with the strong glutamine consumption observed in AMIECs. In this regard, it has been shown in pancreatic cancer cells that glutamine-derived pyruvate plays a significant role in redox homeostasis.

Collectively, our observations and evidence by others suggest the occurrence of two cell-autonomous, primary deficiencies in AMI patient-derived ECs, namely (1) a strong proton coupling in the mitochondrial membrane that results in excess ROS production, countered by enhanced NADPH and glutathione levels, driven by a glycolysis-to-PPP shunt and an enhanced glutamine metabolism; and (2) a switch in carbon source preference from glucose to glutamine, possibly as an adaptive mechanism aimed at potentiating the production of glutathione, while employing alternative pyruvate-to-lactate conversion pathways that replenish NAD+, thus maintaining a viable redox homeostasis (*Kerk et al., 2022*).

Primary dysfunctions in mitochondria, from either inherited or acquired genetic defects, have been suggested as underlying causes of atherogenic pathologies (*Dabravolski et al., 2022*). Alternatively, our AIMECs might continue to carry, ex-vivo, epigenetic imprints of their exposure to factors and cells while in the patients, which could determine altered phenotypes through many cellular generations. Independent of the original insult, a possible intervention to revert the observed mitochondrial dysfunction would be the use of pharmacological uncouplers to mitigate ROS production and the accompanying metabolic reprogramming (*Cadenas, 2018*). Relevantly, knockout of the uncoupling

**Table 1.** Clinical characteristics of patients providing coronary endothelial outgrowth following thrombectomy for ST-segment elevation myocardial infarction.

Values are number (%) or mean ± standard deviation. ACE = angiotensin-converting enzyme; CABG = coronary artery bypass grafting; PCI = percutaneous coronary intervention.

|  | n=8 |
|---|---|
| Age, years | 60±15 |
| Gender, male | 5 (66%) |
| **Medical history and risk factors** | |
| Previous myocardial infarction | 1 (12%) |
| Previous PCI/CABG | 1 (12 %) |
| Current smoker | 3 (37%) |
| Ex-smoker | 2 (25%) |
| Hypertension | 1 (12%) |
| Hyperlipidaemia | 4 (50%) |
| Family history of premature coronary heart disease | 1 (12%) |
| Diabetes mellitus | 0 |
| **Medication on admission** | |
| Aspirin | 2 (25%) |
| Clopidogrel | 1 (12%) |
| B-Blockers | 1 (12%) |
| ACE-Inhibitors | 1 (12%) |
| Statins | 3 (38%) |
| **Myocardial injury** | |
| Troponin I concentration, micrograms/L | 27.9±20.0 |
| **Culprit vessel** | |
| Left anterior descending artery, n=12 | 1 |
| Circumflex artery, n=2 | 1 |
| Right coronary artery, n=23 | 4 |

protein UCP1 caused vascular dysfunction, atherogenesis, and NLRP3 inflammasome activation and IL1-β-dependent inflammation, relieved by re-expression of UCP1 or treatment of animals with chemical uncouplers (*Gu et al., 2021*).

## Methods

### Endothelial cells

AMIECs were isolated from coronary atherothrombotic specimens in patients undergoing percutaneous coronary intervention with thrombectomy for the treatment of acute STEMI at the Royal Infirmary of Edinburgh, Scotland, UK. Relevant clinical features of the patients are summarized in *Table 1*. The study protocol was approved by the Institutional Research Ethics Committee, and all subjects provided written informed consent. Specimens were washed with phosphate-buffered saline (PBS) and manually disaggregated. Tissue explants were seeded onto collagen-I coated six-well plates and maintained under standard cell culture conditions. After 24 hr, tissue explants, non-adherent cells, and debris were aspirated. The medium was changed every other day until the first passage of coronary endothelial outgrowth cells emerged and cells were cultured as described (*Brittan et al., 2015*; *Padfield et al., 2013*; *Tura et al., 2013*), on 0.2% gelatin-coated plates with EGM-2 BulletKit medium (Lonza, Basel, Switzerland), containing EBM-2 basal medium along with the EGM-2 SingleQuots kit components, 10% fetal bovine serum (FBS) and penicillin-streptomycin. Human coronary artery endothelial cells (HCAECs) were purchased from Lonza Clonetics (Walkersville, USA) and used as controls. All cell cultures were maintained in a 37 °C humidified incubator at 5% CO2.

### Cell proliferation and viability assays

Cells were counted either with a Scepter Handheld Automated Cell Counter (Merck Millipore, Billerica, MA, USA) or by Hoechst nuclear staining (HO33342, Sigma-Aldrich) (*Aguilar et al., 2016*).

### In vitro migration (wound healing) assay

Cells were cultured on 12-well plates in a complete medium supplemented with 10% FBS and antibiotics. After reaching confluence, cells were starved for 1 hr in a medium containing 0.5% FBS and mitomycin C (1 mg/mL), a cytostatic drug that reacts covalently with DNA forming crosslinks between the complementary strands of DNA, preventing the separation of the complementary DNA strands and inhibiting DNA replication.

### Tubulogenesis assay

Endothelial tube formation was monitored using an in vitro angiogenesis assay kit (Merck Millipore ECM625) following the manufacturer's instructions. Briefly, EC cells were cultured in 96 wells plates coated with 5% Matrigel (growth factor-free, Corning # 354230) in EGM2 medium without FBS and supplemented with 1% non-essential amino-acids and antibiotics and left in the incubator for 12 hr. Tube branching points were scored in 5 random fields per concentration using the Angiogenesis plug-in of the Image J software.

### Western blotting

70–80% confluent cells were harvested by trypsinization, washed 2 x with cold PBS, pellets resuspended in RIPA buffer (50 mM Tris pH 8.0, 150 mM NaCl, 0.1% SDS, 1% Triton X-100 and 0.5% sodium deoxycholate) supplemented with protease inhibitor cocktail (Sigma-Aldrich) and incubated on ice for 5–10 min. Equal amounts of protein per sample were electrophoresed on 8% or 12% SDS-PAGE and subjected to Western blotting.

### Quantification of metabolites

Glucose, lactate, glutamate, and glutamine were determined by spectrophotometry (COBAS Mira Plus, Horiba ABX) from cell culture media by monitoring the production of NAD(P)H in specific reactions for each metabolite at 340 nm.

### Analysis of polar intracellular metabolites (TCA cycle intermediates and amino acids)

Cells were grown to 70–80% confluence at the required condition for each cellular model and then media were removed, and plates were washed with ice-cold PBS. TCA cycle intermediates were

extracted with the addition of 100% methanol:$H_2O$ (1:1) mixture and scrapping on ice; then, the extracts were sonicated using a titanium probe (VibraCell, Sonics & Materials Inc, three cycles of 5 s), chloroform added to the lysates and tubes placed in a shaker for vigorous agitation at 4 °C for 30 min. Subsequently, samples were centrifuged and the upper aqueous phase was separated and evaporated under airflow at room temperature, adding dichloromethane ($CH_2Cl_2$) for complete dehydration of the samples. TCA cycle intermediates were derivatized by adding 2% (v/v) methoxyamine hydrochloride in pyridine and shaken vigorously at 37 °C for 90 min. Next, N-methyl-N- (tert-butyldimethylsilyl) trifluoroacetamide (MBTSTFA) and 1% tertbutyldimetheylchlorosilane (TBDMCS) were added, samples incubated for 1 hr at 55 °C and finally transferred to GC/MS vials. GC/MS analysis was performed under an electron impact ionization model (*Cascante and Marin, 2008*; *Dettmer et al., 2007*). The relative concentration of the polar intracellular metabolites is quantified using Norvaline (1 mg/mL) as an internal standard.

## Glucose-6-phosphate dehydrogenase activity

Cell extracts were prepared using lysis buffer 20 mM Tris-HCl, supplemented with 1 mM DTT, 1 mM EDTA, 0.02% (v/v) Triton X-100, 0.02% (v/v) sodium deoxycholate and protease inhibition cocktail (Sigma-Aldrich). Lysates were sonicated (VibraCell, Sonics & Materials Inc) and centrifuged at 12,000 × g for 20 min at 4 °C. G6PDH specific activity was measured by adding samples to a cuvette containing 0.5 mM $NADP^+$ in 50 mM Tris-HCl, pH 7.6, at 37 °C. The reaction was initiated by the addition of glucose-6-phosphate at a final concentration of 2 mM (*Aguilar et al., 2016*).

## Intracellular ROS determination

Total intracellular ROS levels were determined by means of flow cytometry using the H2DCFDA probe (Invitrogen). Cells were incubated with incubation buffer 5.5 mM glucose in PBS containing 5 µM $H_2DCFDA$ for 30 min at 37 °C and 5% $CO_2$.

## GSH/GSSG quantification

Fresh cells were lysed with 5% 5-sulfosalicylic acid (Sigma-Aldrich) solution, vortexed, and disrupted by two freeze/thaw cycles in liquid $N_2$ and 37 °C water bath. Fifty µL of this solution was separated for subsequent protein quantification. Cell extracts were kept at 4 °C for 10 min and centrifuged at 10,000 × g for 10 min. For glutathione quantification, 15 U/mL of glutathione reductase and 40 µg/mL of 5,5'-dithiobis(2-nitrobenzoic acid) (Sigma-Aldrich) were dissolved in assay buffer (100 mM $K_2HPO_4$/ $KH_2PO_4$, 1 mM EDTA, pH 7.0).

## Mitochondrial membrane potential

Cells were incubated with MitoProbe DiIC1(5) (1,1',3,3,3',3'-hexamethylindodicarbo -cyanine iodide) with or without CCCP (carbonyl cyanide 3-chlorophenylhydrazone) as a mitochondrial membrane potential disruptor, and scored by flow cytometry.

Detailed protocols are available upon request.

## Acknowledgements

The project leading to this work has received funding from: The European Commission Horizon 2020 research and innovation program under the MOGLYNET H2020-MSCA-ITN-EJD grant (agreement No 675527); CIBER- Carlos III National Institute of Health, Spain (TT and MCas: CIBEREHD-CB17/04/00023; OTC: CIBERES-CP17/00114); Spanish Ministerio de Economia y Competitividad (MINECO) and Ministerio de Ciencia e Innovación -European Commission FEDER funds—'Una manera de hacer Europa' (TT: PID2019-107139RB-C21 and MCas: PID2020-115051RB-I00); Generalitat de Catalunya-AGAUR (MCas: ICREA Academia Prize 2021, MCas: 2021 SGR00350; TT: 2021 SGR1490); Plataforma Temática Interdisciplinary – Salud Global (TT: SGL2103019); NLM is supported by the British Heart Foundation (RE/18/5/34216, RG/20/10/34966, CH/F/21/90010).

## Additional information

### Funding

| Funder | Grant reference number | Author |
|---|---|---|
| CIBER Carlos III National Institute of Health | CIBEREHD-CB17/04/00023 | Timothy M Thomson Marta Cascante |
| CIBER Carlos III National Institute of Health | CIBERES-CP17/00114 | Olga Tura-Ceide |
| Spanish Ministerio de Economia y Competitividad | PID2019-107139RB-C21 | Timothy M Thomson |
| Spanish Ministerio de Economia y Competitividad | PID2020-115051RB-I00 | Marta Cascante |
| Generalitat de Catalunya-AGAUR | 2021 SGR00350 | Marta Cascante |
| Generalitat de Catalunya-AGAUR | 2021 SGR1490 | Timothy M Thomson |
| British Heart Foundation | RE/18/5/34216 | Nicholas L Mills |
| British Heart Foundation | RG/20/10/34966 | Nicholas L Mills |
| British Heart Foundation | CH/F/21/90010 | Nicholas L Mills |
| H2020 Marie Skłodowska-Curie Actions | MOGLYNET H2020-MSCA-ITN-EJD grant (agreement No 675527) | Marta Cascante Timothy M Thomson Erika Zodda Marina Carini Olga Tura-Ceide |
| Catalan Institution for Research and Advanced Studies | ICREA Academia Prize 2021, 2014SGR1017 | Marta Cascante |
| Plataforma Temática Interdisciplinar - Salud Global | SGL2103019 | Erika Zodda Timothy M Thomson |
| Ministerio de Ciencia e Innovación | | Erika Zodda Timothy M Thomson |

The funders had no role in study design, data collection and interpretation, or the decision to submit the work for publication.

### Author contributions

Erika Zodda, Conceptualization, Data curation, Formal analysis, Validation, Investigation, Visualization, Methodology, Writing – original draft, Writing – review and editing; Olga Tura-Ceide, Investigation, Methodology, Writing – review and editing; Nicholas L Mills, Conceptualization, Resources, Funding acquisition, Visualization, Methodology, Writing – review and editing; Josep Tarragó-Celada, Formal analysis, Investigation, Methodology, Writing – review and editing; Marina Carini, Conceptualization, Funding acquisition, Writing – review and editing; Timothy M Thomson, Conceptualization, Resources, Data curation, Formal analysis, Supervision, Funding acquisition, Validation, Investigation, Visualization, Methodology, Writing – original draft, Project administration, Writing – review and editing; Marta Cascante, Conceptualization, Formal analysis, Supervision, Funding acquisition, Validation, Visualization, Writing – original draft, Project administration, Writing – review and editing

### Author ORCIDs

Erika Zodda ⓘ https://orcid.org/0000-0002-4536-9106
Timothy M Thomson ⓘ https://orcid.org/0000-0002-4670-9440
Marta Cascante ⓘ https://orcid.org/0000-0002-2062-4633

## Ethics

Acute myocardial infection patient-derived endothelial cells (AMIECs) were isolated from coronary atherothrombotic specimens in patients undergoing percutaneous coronary intervention with thrombectomy for the treatment of acute ST-segment elevation myocardial infarction (STEMI) at the Royal Infirmary of Edinburgh, Scotland, UK. The study protocol was approved by the Institutional Research Ethics Committee, and all subjects provided written informed consent.

## Decision letter and Author response

Decision letter https://doi.org/10.7554/eLife.86260.sa1

## Additional files

### Supplementary files
• MDAR checklist

• Reporting standard 1.

### Data availability

All data generated or analysed during this study are included in the manuscript.

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
