## [Editor Report]

In a novel and important approach, the authors have generated coronary artery endothelial cells from thrombectomy specimens of patients with acute myocardial infarctions (AMIECs). Compared to control healthy coronary artery endothelial cells (HCAECs) they display solid evidence for impaired growth, migration and tubulogenesis and major metabolic differences. This advances the field and provides opportunities for future studies including investigating new avenues of therapy for AMI.

---

## [Decision Letter]

**Decision letter after peer review:**

Congratulations, we are pleased to inform you that your article, "Endothelial dysfunction in acute myocardial infarction is characterized by cell-autonomous metabolic reprogramming and oxidative stress", has been accepted for publication in *eLife*.

*Reviewer #1 (Public Review):*

Summary of objectives and findings

In a novel approach the authors have generated coronary artery endothelial cells from thrombectomy specimens of patients with acute myocardial infarctions (AMIECs). Compared to control healthy coronary artery endothelial cells (HCAECs) they display impaired growth, migration and tubulogenesis and major metabolic differences.

In a logical sequence of elegant experiments they have shown that AMIECs display augmented reactive oxygen species (ROS) and glutathione intracellular content with low glucose consumption and high lactate production.

With diminished glycolysis there is shunting towards the pentose phosphate pathway (PPP) and NADPH generation compared with glycolysis in HAECs. The glutaminolytic enzyme GLS was upregulated in AMIECs and explained their high glutamine content. The AMIECs showed a significantly higher mitochondrial membrane potential than the control HCAECs.

They conclude that high mitochondrial proton coupling underlies the abnormally high production of ROS, balanced by PPP- and glutaminolysis-driven synthesis of glutathione, as a primary, cell-autonomous abnormality driving endothelial dysfunction in AMI.

Strengths and weaknesses

The abstract has a very clear and comprehensive summary of the background, methods, results and conclusions. The main body of the manuscript is very well written with an explanation of the context together with results and interpretation for each sub-study and with clear and convincing illustrations for each sub-study. This makes it very easy to read. Strengths include the development of the method to study AMIECs which is a major advance. A weakness is the use of only one batch of commercial cells as a control which may limit the conclusions reached. Another issue is whether the ex vivo AMIECs reflect what happens in vivo. This and potential implications could be discussed.

*Reviewer #3 (Public Review):*

The study provides detailed enzymatic and metabolic analysis of endothelial cells(ECs) derived from myocardial infarction patients and attempts to link these metabolic changes to the slow migration and tubulogenesis phenotype of patient cells. The study it is of great interest to the field of cardiovascular biology and provides the readers with insights into plausible areas of therapeutic interventions to restore functionality of ECs. The study also goes ahead and establishes a cell culture model for 9 patient derived EC lines. The authors demonstrate that AMIECs have reduced glycolytic rates owing to shunting of glucose into the pentose phosphate pathway leading to heightened NADPH production. The NADPH was suggested to be linked to balancing the heightened ROS levels in AMIECs. The authors then go on to demonstrate that AMIECs utilize glutamine as a major carbon source and via its conversion to glutamate, provides for glutathione synthesis.

The study holds immense merit, however, the authors have missed out on going back and attempting to actually show that the metabolic differences play a major role in contributing to the migration/tubulogenesis defects observed in patient cells, making the study rather descriptive in nature. By attempting to address this concern, the authors will be able to demonstrate that by modulating a few metabolic fluxes in patient endothelial cells, functionality can be restored -- bringing forth the true potential of this very important work. Also, this is an ex-vivo system so the results need to be treated as such. However, this does not discount the crucial findings of the study,

Additional comments

The authors have succeeded in producing AMIECs to study and compare with HAECs. This advances the field and provides opportunities for future studies including investigating new avenues of therapy for AMI.

*Reviewer #1 (Recommendations for the authors):*

On page 14 under Methods Endothelial cells – infection should be corrected to infarction

Please include/clarify how the cells were transported from Edinburgh to Spain and at what stage. Were any of the experiments done in Edinburgh?

*Reviewer #3 (Recommendations for the authors):*

Specific comments:

1. It is not clear how the N=3 was achieved if each explant could only be cultured for a certain amount of time and likely only one biopsy sample was obtained from each patient.

2. In figure 1B, 1C the authors are advised to check the purity of the ECs by Flow cytometry. From the methods, it appears that the cells were derived via explants. To ensure purity, Authors must check for common confounding cell types via Flow cytometry.

3. Figure numbers are not correct in Figure 1. Figure1D, 1E, 1F need to be checked/corrected.

4. Figure 3A: Perhaps a better way to represent this data will be to show Lactate and glucose on two separate graphs. The current graph can be very misleading if one does not read the legend. This will be important for the overall improvement of data representation in the paper.

5. Figure 4: The authors need to treat the cells with a ROS scavenger and check the glutathione levels to positively make the statement they have made about the ROS-Glutathione balance.

6. The Authors must use inhibitors for Glutaminsase activity and check whether the treatment affects the downstream observations relating to anaplerotic reactions or not. At this point the evidence is mostly extrapolative without confirmatory experiments.

7. The paper will greatly benefit from simple carbon tracing experiments to show the flow of metabolic fluxes in AMIECs vs controls. The experiments will be conclusive in nature and of great value.

8. The authors need to utilize drugs to modulate the identified metabolic points- such as PPP /glutamine utilization/ Enhancing glucose uptake/ inhibiting ROS etc. Following such manipulations, the authors must check for phenotypic changes in AMIECs. In case the claim the authors make is correct- that the metabolic differences largely are responsible for the slower migration, the enzymatic manipulations should alter the phenotype towards normalcy.